# Effectiveness of New Rock-Ramp Fishway at Miyanaka Intake Dam Compared with Existing Large and Small Stair-Type Fishways

**Taku Masumoto [1,*], Masahiko Nakai [2], Takashi Asaeda [3] and Mizanur Rahman [3]**

1   Energy Planning Department, East Japan Railway Company, Tokyo 1518578, Japan
2   Japan International Consultants for Transportation Co., Ltd., Tokyo 1000005, Japan; nakai@jictransport.co.jp
3   Graduate School of Science and Engineering, Saitama University, Saitama 3388570, Japan; asaeda@mail.saitama-u.ac.jp (T.A.); masudbiochem2012@gmail.com (M.R.)
*   Correspondence: t-masumoto@jreast.co.jp

**Abstract:** The migration of fish is influenced by the unique environmental characteristics of the destination and migratory habitat preferences. There are three fishways in Miyanaka Intake Dam. The rock-ramp fishway was newly established in 2012, creating an environment with different flow velocities and water depths. The purpose of this study was to investigate the effectiveness of the new rock-ramp fishway for native fish through two surveys. In the first survey, traps were installed during the survey period in all three fishways and all fish were caught. The run-up environment was quantified by measuring the flow velocity. In the second survey, fish were caught by spectrum methods upstream and downstream from the dam. It was found that bottom-dwelling fish and swimming fish not bound to the bottom with low migration abilities used the rock-ramp fishway for migration and as a habitat. After the new rock-ramp fishway was built, catches increased upstream from the dam. Further, the rock-ramp fishway is a potential habitat for certain species, such as *Cottus pollux*. As this is the first study to demonstrate the effectiveness of rock-ramp fishways, the research results are expected to be valuable to fishery managers and those planning river engineering projects.

**Keywords:** rock-ramp fishway; bottom-dwelling fish; fish biodiversity; river continuity; diadromous migratory fish; *Plecoglossus altivelis*



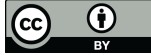

## 1. Introduction

Although river channel continuity is essential for fish migration, it is often disrupted by dams and weirs [1–5]. Dams and other human-made barriers create obstacles in aquatic ecosystems by altering habitats and fragmenting the lotic system [6]. Restoring continuity is therefore particularly important for rivers disrupted by dams [4,7–12]. Fishways have been developed for this purpose [4,13–18], and the effects of fishways on large varieties of fish have been tested vis adaptive management based on biological, hydraulic, and other physical parameters [19–22]. Further, measures to attract fish to the fishways are being studied [23]. The main targets of such studies have been anadromous salmonids, such as the chum salmon *Oncorhynchus keta*, brown trout *Salmo trutta*, Chinook salmon *Oncorhynchus tshawytscha*, and rainbow trout *Oncorhynchus mykiss* [24,25] because these species are globally important as aquatic products [7]. However, the conservation of endangered fish species is also important [8], and freshwater fish, such as *Plecoglossus altivelis*, *Tribolodon hakonensis*, and *Cottus pollux*, are also valuable species in the fishing industry as equivalents to the anadromous salmonids in Japan.

Several types of technical fish passes have been proposed and installed in rivers worldwide, including the vertical slot and Denil fishways [9]. New styles such as rock-weir fishways have also been proposed considering the volumetric dissipated power and other factors [26,27]. In the Columbia Snake River system, ice-harbor fishways are beneficial for

adult salmonids [28]. A rock-ramp fishway is similar to a natural fishway because the flow is turbulent. Fish are usually attracted to turbulent flows if their stability mechanisms are adequate for the given hydrodynamic environment and if the flow has expected spatially secure qualities [29–31]. If the flow volume is large, the velocity inside the fish pass becomes considerably high for many fish in addition to the large fish species such as *O. keta* [32–41]. Except for rock-ramp and bypass fishways, most fish passes, have concrete walls and bottoms with panels to reduce the flow velocity inside the fish pass. Rock-ramp fishways have reduced gradients to further reduce the flow velocity compared to larger and smaller fishways due to the formation of meandering waterways. Another feature is that many normal fishways do not have riverbed material at the bottom. However, riverbed materials are essential for bottom-dwelling fish species and other benthic organisms [7,42–46]. Thus, installing other types of fishways with gentle slopes and riverbed materials is crucial for preserving the biodiversity of the river channel [3,10].

The fishway of Miyanaka Intake Dam was improved in March 2012 to ensure continuity between the flow through the fishway and the main flow downstream from the dam. Furthermore, considering fish biodiversity, a new rock-ramp fishway was established to facilitate migration and provide a habitat for the bottom-dwelling fish. The rock-ramp fishway is called the "Seseragi Fishway". This initiative is also important for diadromous migratory fish, such as *C. pollux*, which are bottom-dwelling fish. Before these improvements, only two fishways, namely large and small stair-type fishways, were installed; thus, the conditions were not favorable for upstream migration of the bottom-dwelling fish. No previous studies have investigated the effectiveness of rock-ramp fishways.

In this study, we investigated whether the newly constructed rock-ramp fishway could achieve this objective. This paper reports the results of the corresponding study and demonstrates the effectiveness of the new fishway. The combined results of the survey conducted on the fishway of the dam and those of the surveys conducted upstream and downstream of the dam confirm the effects of the newly established rock-ramp fishway.

## 2. Materials and Methods

### 2.1. Study Area

The Shinano River starts from a mountainous zone at an elevation of approximately 2160 m, flows over 367 km, and discharges into the Japanese Sea at Niigata. The riverine habitat contains typical features, such as scour pools, riffles, and runs, which have generally been well preserved as unit structures. Miyanaka Intake Dam is located at the midstream of the river, 134 km upstream from the river mouth. In the upstream and midstream reaches of rivers located near mountainous areas, the surface of the river channel bed is often composed of gravel and bed load [47–50]. A relatively large amount of sediment is generated upstream of the Miyanaka Intake Dam because of the floods. The transport of gravel and quicksand is affected by riverbed material [51]. The rate of material transport deposited on the riverbed depends on the size of the particles in the water. Therefore, the riverbed material of the Shinano River middle basin around the Miyanaka Intake Dam is composed of gravel with a diameter of 100 to 150 mm.

Miyanaka Intake Dam (37°3′58.445″ N, 138°41′50.321″ E) was built in 1939, and the fishway installed since the time of construction was improved once in 1986. Two sizes of stair-type fishways—the first measuring 218 m long and 10 m wide and the second 262 m long and 2 m wide, folded 110 m from the upstream entrance—were installed to compensate for an altitude difference of 6 m between the upstream and downstream water levels across the dam. However, they were insufficient for fish migration in the river and thus reconfigured in 2012 to target a wider range of species [52]. Figure 1a–c shows the location of Miyanaka Intake Dam, the survey points utilized in this study, and the fishway.

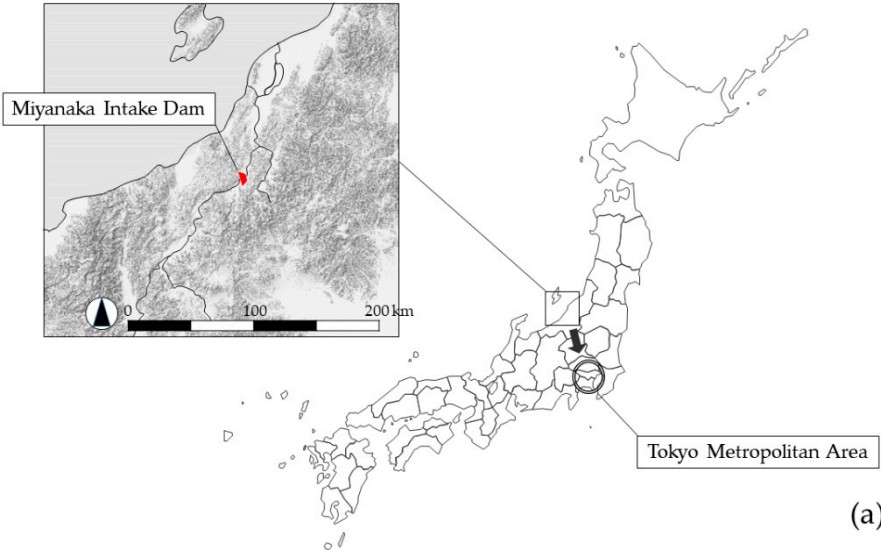

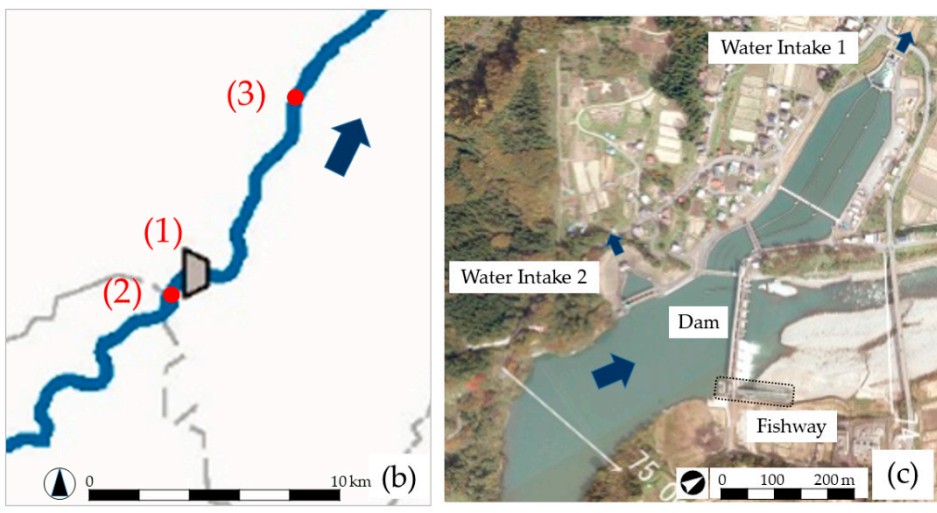

**Figure 1.** (**a**) Location of Miyanaka Intake Dam, Japan. (**b**) Three survey points utilized in this study: (1) Miyanaka Intake Dam fishway, (2) a point upstream from the dam (Himizo, Tokamachi City), and (3) a point downstream from the dam (Tokamachi Bridge). Because only the maintenance flow rate of 40 m$^3$/s is discharged downstream from the dam, the river's discharge rate and width are lower in (3) than (2). However, that has not changed since 2012. In addition, (2) was not affected as a flooded area because it was set further upstream of the flooded pond. (**c**) General overview of the fishway, which is located on the right-side bank of the dam, and water intake is located on the left-side bank. The maintenance flow rate (40 m$^3$/s) is discharged from the gate on the right bank side, where the fishway is located, to exert the priming effect on the fishway. The maximum water intake is 166.96 m$^3$/s for water intake 1 and 150 m$^3$/s for water intake 2. Water is sent from the water intake to the power plant through a water tunnel.

*2.2. Fishway Design*

The most severe problem with the Miyanaka Intake Dam fishway was the generation of shear waves. Until 2009, the maintenance flow rate discharged from Miyanaka Intake Dam was 7 m$^3$/s and increased to 40 m$^3$/s in 2010. As a result, it has become possible to determine the flow rate of the fishway appropriately.

The maximum flow velocity was 1.31 m/s for the general part of the large fishway, 1.66 m/s for the notch, and 1.12 m/s for the small fishway until 2011 (related Table 1). As

the maximum flow velocity exceeded the burst speeds of *P. altivelis* and *T. hakonensis*, it was necessary to reduce the flow velocity of the fishway, which required reducing the discharge.

**Table 1.** Fishway specifications before improvement (from 1986 to 2011). Two fishways with different water depths and flow velocities were constructed at Miyanaka Intake Dam. The details are shown in Figure 2a.

| Fishway Type | | Gap (m) | Water Depth (m) | Width (m) | Velocity (m/s) | Discharge (m³/s) |
|---|---|---|---|---|---|---|
| Large fishway | General part | 0.25 | 0.25 | 10.0 | 1.31 | 3.7 |
| | Notch part | | 0.39 | | 1.66 | |
| Small-fishway | – | 0.25 | 0.19 | 2.0 | 1.12 | 0.3 |

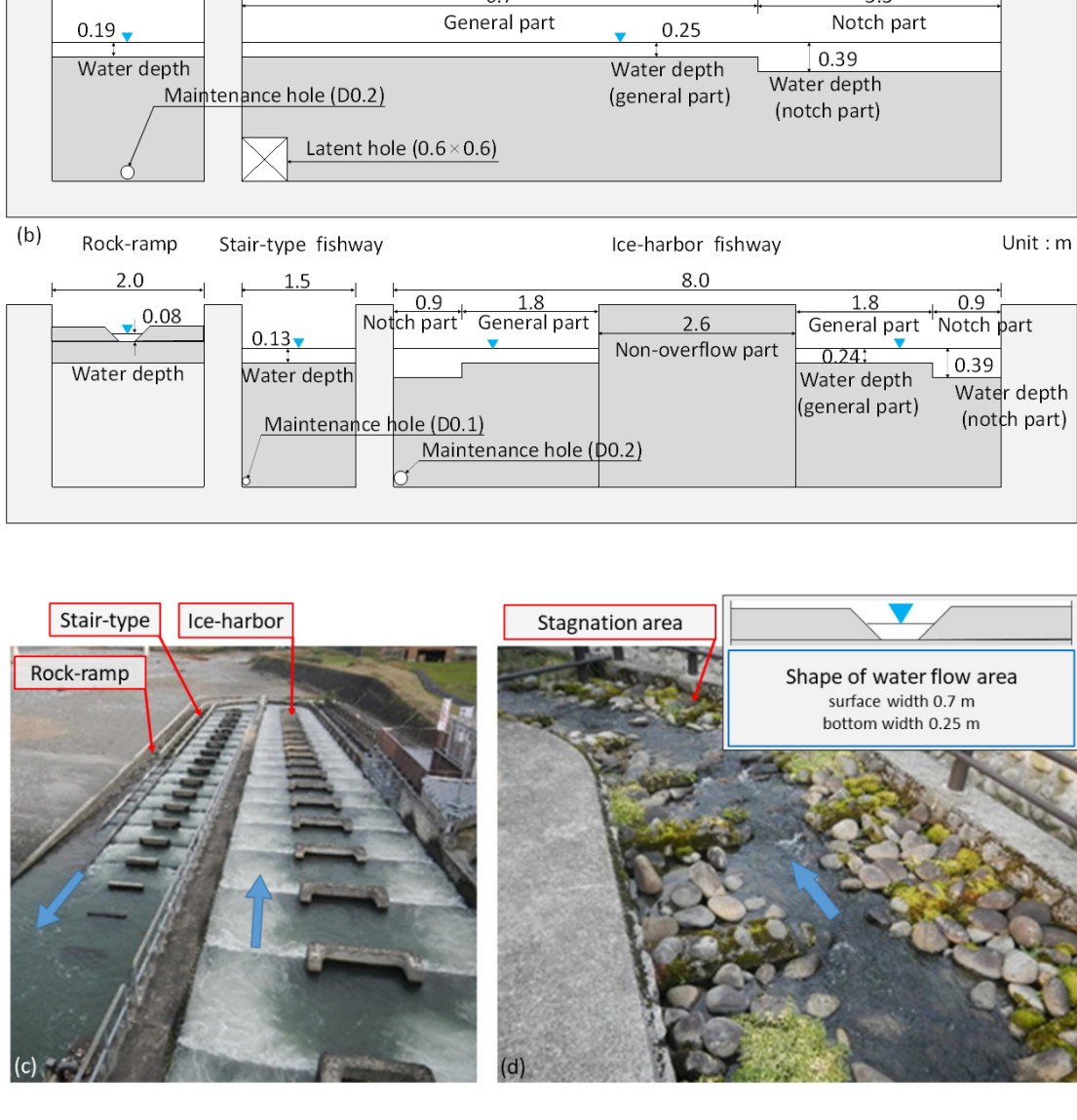

**Figure 2.** Fishway design (**a**) before and (**b**) after improvements, as seen from the downstream side. (**c**) View of all three fishways (ice-harbor-, stair-, and rock-ramp-type fishways) from the upstream side. (**d**) Rock-ramp fishway; the trapezoidal cross-section contributes to the provision of various habitats.

Sixteen target fish species were identified to determine the necessary flow velocity and water depth for the fishway in the design process set up with favorable conditions. The expected fish species were determined for each type of fishway. The ice-harbor fishway targeted *O. keta*, *Oncorhynchus masou*, and *Oncorhynchus mykiss*; the stair-type fishway targeted *P. altivelis*, *T. hakonensis*, *Tribolodon nakamurai*, and *Cyprinus carpio*; and the rock-ramp fishway targeted *Liobagrus reini*, *C. pollux*, and *Rhinogobius kurodai*.

A suitable overflow depth was determined through field experiments so that fish could easily swim and hydraulic conditions under which transverse waves could be fully suppressed.

The measures employed to suppress transverse waves are as follows:

1. The width of the large fishway was decreased from 10 m to 8 m.
2. The large fishway was changed to an ice-harbor fishway.
3. The latent holes that caused transverse waves were abolished, and small holes (0.2 m diameter) were arranged at regular intervals for maintenance.
4. The notch width was decreased.

The water depth was set to 0.24 m for the general part of the ice harbor, 0.39 m for the notch, 0.13 m for the stair-type fishway, and 0.08–0.15 m for the rock-ramp fishway which is suitable to suppress the generation of transverse waves while maintaining the falling flow state shown in Figure 2b [34,35,53]. The depths of the ice-harbor and stair-type fishways were measured at the top of the wall of the fishway. The water depth of the rock-ramp fishway was measured at the central part of the upstream end. The measurements confirmed that the target water depth was achieved. To ensure this run-up environment, it is necessary for the water level of the fishway to follow the water level changes of the dam. Therefore, the gate of the fishway was modified so that it would change automatically according to the water level of the dam.

Based on these studies, the Miyanaka Intake Dam fishway was modified in March 2012. Figure 2a shows the specifications of the fishway before the improvements, and Figure 2b–d illustrates the specifications after the improvements. The large fishway was changed from stair type to ice-harbor type without changing the terrain characteristics, such as altitude and plane positions of the entrance and exit of each fishway, through both numerical analyses and field release experiments [54]. To suppress fluctuations in the water surface, the folded part was improved to a semi-arc shape [55].

The structure of the outer wall of the fishway was not changed to minimize the scale of structural improvement. The old small fishway was covered with concrete, and a new rock-ramp fishway was built over it. Therefore, the stair-type fishway was installed in a space in which a part of the large fishway was partitioned by a wall. As there were no problems with the flow conditions of the small fishway, the small fishway was relocated, with its width decreased from 2 m to 1.5 m while maintaining the stair-type structure.

The folded structure of the fishway was unaltered because it did not affect the passage rate and was therefore deemed suitable for fish passage [56,57]. A meandering rock-ramp fishway of length 257 m and width of 2 m, was then constructed around the existing ones, with a pool type (stair-type) structure. The rock-ramp fishway was established to improve the environment where bottom-dwelling fish and fish with low swimming abilities can move and inhabit [58]. To reproduce conditions similar to a natural mountain stream and various water areas, natural stones were placed to form a pool. The maim flow meandered, and the gradient was relaxed from 6.7% to 5% to accommodate various aquatic organisms, such as bottom-dwelling fish and fish not bound to the bottom with low swimming abilities. Therefore, the distance between the pool bulkheads (length in the extension direction) was set to 1 m, and the water drops between pools were set to 0.05 m. Unfixed cobblestone with a diameter of 0.15 m was laid in the waterway of the fishway. Thus, it was possible to provide a stagnant area as a place for rest and habitat by maintaining the state of the floating stones.

With ice-harbor fishways, large latent holes are usually created to maintain the force of the falling water. An additional improvement was made to the ice-harbor fishway at the back of the non-overflow area stationary such that it acts as a rest area for fish moving upstream [59–61]. The overflow depth, discharge, and width of the rock-ramp fishway were 0.08 m, 0.022 $m^3$/s,

and 2 m, respectively. Meanwhile, the cross-section was trapezoidal, and the water surface width and bottom width of the channel were 0.7 m and 0.25 m, respectively. To form water areas with various hydraulic quantities (water depth and flow velocity) in the overflow part, natural stones were used for the partition wall. In addition, to create a fishway emulating a natural river with repeating rapids and deeper water, a backwater zone was established, and loose cobbles were used. Consequently, various flow conditions occurred in the fishway, and bottom-dwelling fish, fish with low swimming abilities, and crustaceans were able to move and inhabit the area. The side walls at the fishway entrances were cut down so that fishways with different flow velocities could be selected according to swimming abilities after the fish invade the fishway. This makes it easier for bottom-dwelling fish and fish with low swimming abilities to enter the rock-ramp fishway.

*2.3. Data Collection*

2.3.1. Quantification of the Fishway Run-Up Environment

Velocity distribution measurements were previously obtained using floats and aerial images [43,62–65]. The measurement position of the ice-harbor fishway, the stair-type fishway, and rock-ramp fishway flow velocity were set at a depth of approximately 0.1 m, 0.05 m, and 0.05 m, respectively, from the water surface at the center of the separating wall. The flow velocity was measured with a handheld electromagnetic current meter after the measured value had stabilized at the measurement position. Each measurement was performed twice (unit: m/s, with a resolution up to the second decimal place), and the average value (rounded to the third decimal place) was adopted. For the measurements, the TK-105X device from Toho Denso was used.

Relationships between turbidity and fish habitat [66–71] and river water temperature and fish habitat [72,73] have been reported. Turbidity and river water temperature are considerably affected by rainfall and snowmelt upstream of the dam and are also related to discharge from the dam gate. Therefore, prior to examining the effectiveness of the fishway and the effects of the rock-ramp fishway, the relationship between the number of captured fish and environmental factors was confirmed. To confirm the relationship between the numbers of *P. altivelis* and *T. hakonensis* that had large run-up numbers during this period and the environmental factors, the fishway water temperature and suspended solids (SS) were measured, and the discharge from the gate was recorded. As it was thought that the run-up increased with increasing water temperature, the water temperature was continuously measured by a data logger installed in the fishway. Because it was previously reported that *P. altivelis* were repelled when the SS content exceeded 22 mg/L [74], the SS content was measured once daily by collecting water from the fishway. When the discharge from the gate was high, it was considered that the run-up of the fish was small; thus, the average daily discharge from the gate was recorded. Next, it was confirmed by frequency distribution in SPSS performing one-way ANOVA that environmental factors did not significantly affect the survey results.

2.3.2. Survey for Capturing Fish

In the fishways, only trap-based surveys were conducted at the upstream end. Figure 3 shows the fishway monitoring status. The traps were installed during the survey period at the upstream ends of all fishways. To cover the entire widths of the fishways, six traps were installed on the ice-harbor fishway, two traps were installed on the stair-type fishway, and one trap was installed on the rock-ramp fishway. The trap consisted of an iron frame and a net basket. On the downstream side (entrance for fish) of the net basket, a guide part that narrows toward the upstream (entrance of the net basket) was provided. After passing through the induction part, the fish entered the net basket from the entrance of the net basket. At the S-1 point of the rock-ramp fishway, a net basket was installed in the main flow to capture all fish. Therefore, a partition net was installed so that the fish could migrate only in the main flow. The fish were then collected in the opening of the net basket. During the survey period, the basket was manually retrieved every hour from the water to prevent the fish from filling up the net basket and clogging the mesh of the net basket with dust, and all the fish in the basket were

caught. A survey to count the species and individuals of fish was conducted from 6 June to 4 July, 2012, 2014, and 2015, and from 6 June to 10 July 2013 [75]. Prior to this survey, a 24 h survey including nighttime was conducted for two days, from 8 to 9 June 2010. However, most fish were not caught at night. Thus, the survey from 2012 was conducted from 7:00 to 19:00. For all trapped fish, photographs were captured, and the body lengths and weights were measured before the fish were released back into the river.

In the rock-ramp fishway, the survey was conducted not only at the upstream end (S-1) but also at midpoints (80 m (S-2), 145 m (S-3), and 210 m (S-4) from the upstream end) to determine the fish inhabiting the rock-ramp fishway. The fish were then collected in the opening of the catch basket (height 0.2 m, width 0.2 m). This survey was conducted three times a day (at 8:00, 13:00, and 18:00). The unfixed boulders were moved, and the landing nets were used to capture the bottom-dwelling fish and small fish with weak migration abilities. No electric shocker was used in this study to compare with past findings. The captured fish were counted, their lengths were measured, and they were released upstream from the dam.

The results of investigations conducted upstream and downstream from Miyanaka Intake Dam were used to confirm the effects of the new rock-ramp fishway. At both the upstream and downstream locations, surveys of the fish caught were performed over two days three times annually at the ends of June, August, and October. The surveys were conducted by the Ministry of Land, Infrastructure, Transport, and Tourism and East Japan Railway Company before 2012 when the Miyanaka Intake Dam fishway was improved. The upstream survey site, Himizo in Tokamachi City, was 1.5 km upstream from the dam. The downstream survey site, Tokamachi Bridge, was 10 km downstream from the dam (Figure 1b). Each survey reach was 1 km long where rapids, flats, pools, and wandoes exist. Fish were caught by the methods shown in Table 2. Cast net is fishing gear suitable for shallow waters, such as flats. The target fish are large-migratory fish, such as *P. altivelis* and *T. hakonensis*, and large bottom-dwelling fish. Landing nets are suitable fishing gear for places such as riverbank vegetation zones, submerged vegetation zones, riverbed stones, sand, and mud, where the target fish are small fish and fry. Stationary nets are fishing gear used in places where the depth can be fixed with a weight or secured with a stake and places that can be paths for fish. A bag net was placed on the upstream side, and one of the sleeve nets was affixed at the riverbank. A place with a low flow velocity was selected as the position to stretch the sleeve net, where the target fish are of all types including nocturnal fish. Gill nets are used in places where the flow is calm and fish can pass; here, the target fish are of all types, including migratory and nocturnal fish. Longline fishing gear is used near obstacles and in deep water, where the target fish are nocturnal carnivorous fish, such as *Silurus asotus* and freshwater *Salvelinus leucomaenis pluvius* and *O. masou*. The caught fish were counted, and their maximum and minimum lengths were measured. The fish that could not be identified at the site were brought to the laboratory and identified in detail.

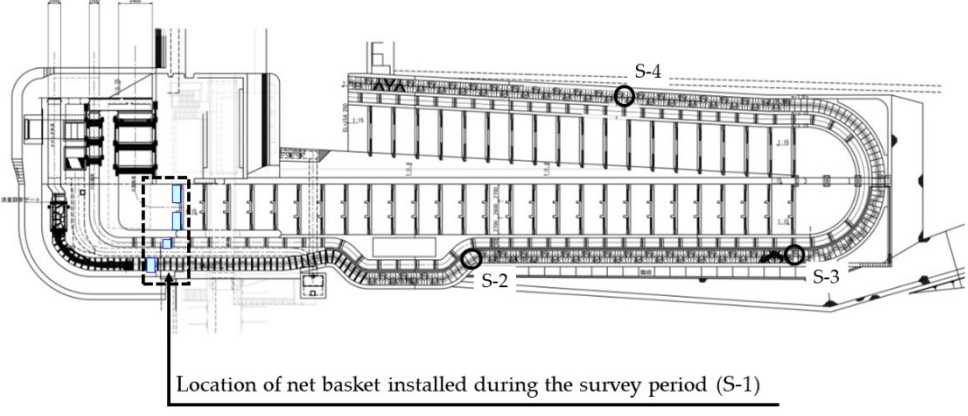

(a) A panoramic view of the survey point

**Figure 3.** *Cont.*

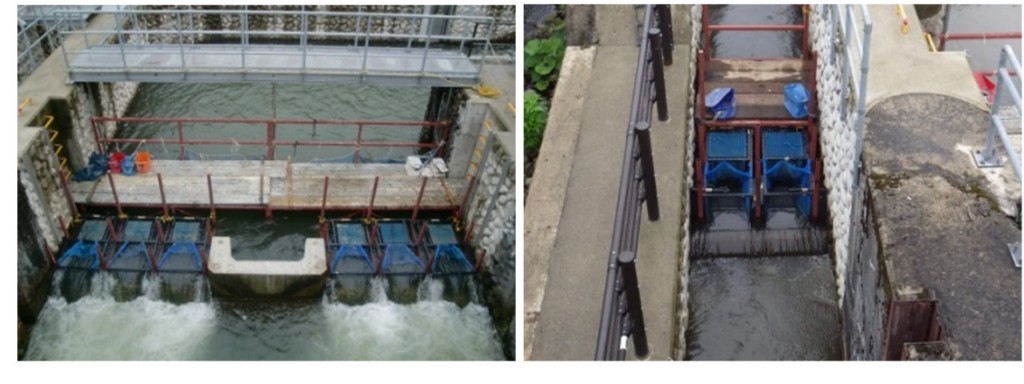

S-1(ice-harbor)          S-1(stair-type)

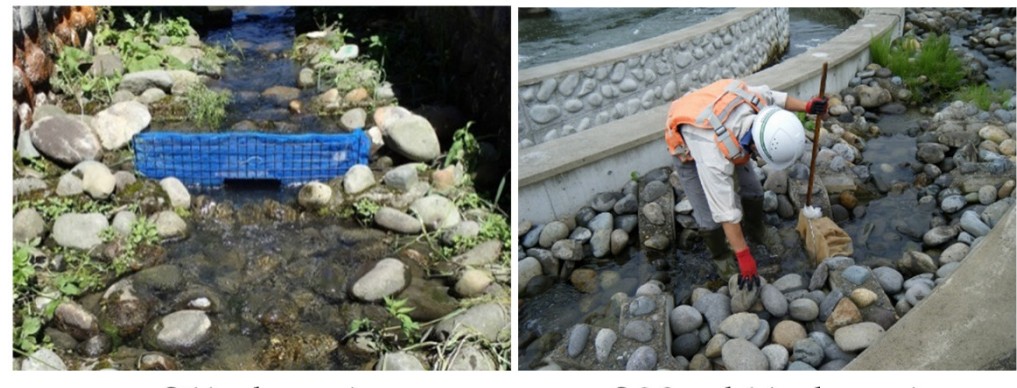

S-1(rock-ramp)          S-2,3 and 4 (rock-ramp)

(b) Status of each survey point

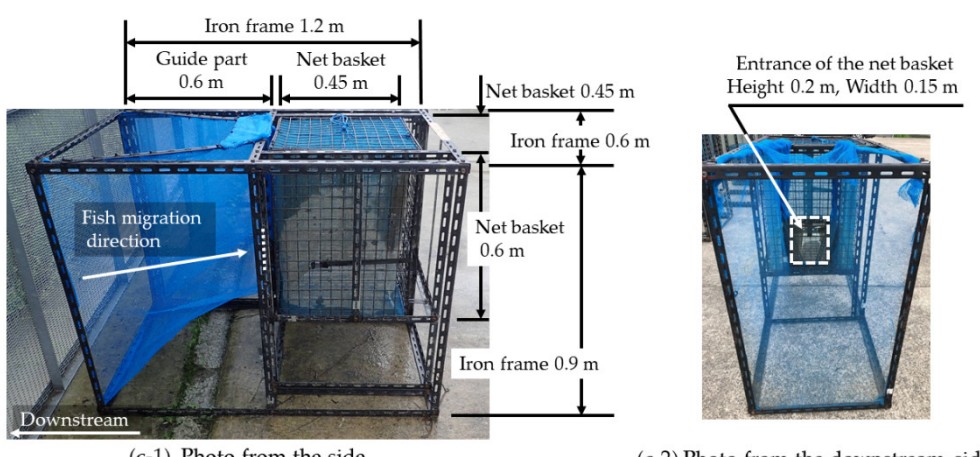

(c-1) Photo from the side          (c-2) Photo from the downstream side

(c) S-1 Trap (ice-harbor and stair-type fishway)
(Iron frame and Net basket with a mesh of 2 mm)

**Figure 3.** *Cont.*

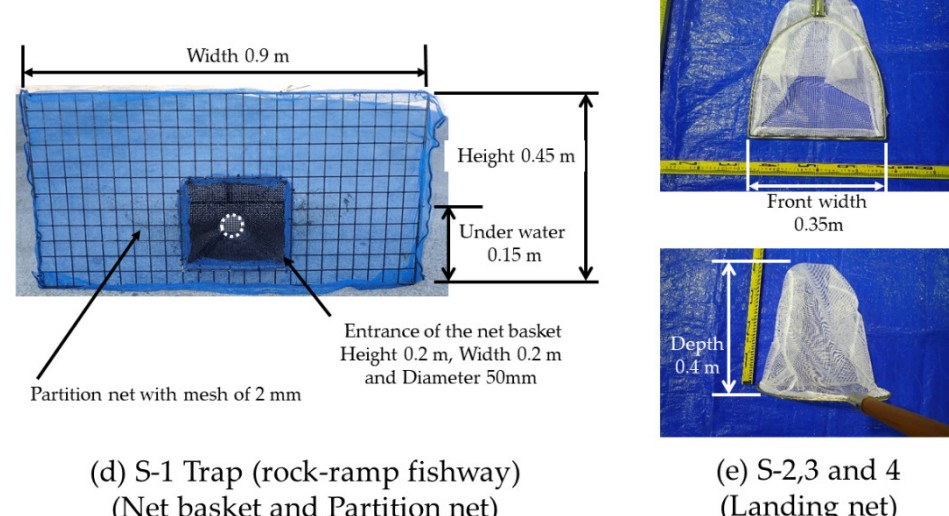

(d) S-1 Trap (rock-ramp fishway)
(Net basket and Partition net)

(e) S-2,3 and 4
(Landing net)

**Figure 3.** Fishway monitoring status. (**a**) The traps were installed at the upstream ends (S-1) of the ice-harbor, stair-type and rock-ramp fishways. The survey points in the rock-ramp fishway were set at not only the upstream end but also the midpoints. (**b**) The number and widths of the traps were adjusted according to the fishway's width to prevent gaps between the traps. One trap was installed on the rock-ramp fishway, and no traps were installed at the midpoints of the rock-ramp fishway. (**c**) The traps installed in the ice-harbor and stair-type fishways consisted of iron frames fixed to the fishway during the survey and net baskets. (c-1) On the downstream side of the net basket, the guide part was installed. (c-2) At the connection between the net basket and guide part, the entrance to the fish was prepared. (**d**) The trap installed at the upstream end survey point of the rock-ramp fishway comprised a small net basket commonly used to catch sculpins. (**e**) Landing nets were used in the survey of the midpoints (S-2, 3, and 4) of the rock-ramp fishway.

**Table 2.** Survey methods upstream and downstream from the dam. The fishing gear, specifications, and amount of effort were based on the River Waterside Census conducted by the MLIT nationwide. To catch various fish, the surveys were conducted according to the habitat of the survey location.

| Fishing Gear | Specification | Amount of Effort | Survey Habitat | | | |
|---|---|---|---|---|---|---|
| | | | Rapid | Flat | Pool | Wando |
| Cast net (mesh size 12 mm) | Circumferential length of the hem of the net: 12 m | [40 times] | 10 times | 10 times | 10 times | 10 times |
| Cast net (mesh size 18 mm) | Circumferential length of the hem of the net: 15 m | [40 times] | 10 times | 10 times | 10 times | 10 times |
| Landing net | Caliber 0.3 m, mesh size 2 mm and with rock movement | [4 h] | 1 h | 1 h | 1 h | 1 h |
| Stationary net | Bag net; mesh size 10 mm, diameter 0.5 m, length 7 m Sleeve net; mesh size 14 mm, height 1.2 m, length 2.5 m, with float | [2 night] From before sunset to after sunrise | – | 1 night | – | 1 night |
| Gill net | Length 20 m, height 1.2 m, mesh size 15 mm, and 90 mm, with two gill nets being used side by side in one survey habitat | [2 night] From before sunset to after sunrise | – | 1 night | 1 night | – |
| Longline | Five needles per 1, and the bait is mainly earthworms | [4 night] From before sunset to after sunrise | 1 night | 1 night | 1 night | 1 night |

## 3. Results

### 3.1. Quantification of the Fishway Run-Up Environment

Because the gate was improved so that the flow rate could be maintained stably, the flow velocity was stable (Table 3).

**Table 3.** Fishway specifications. Three fishways with different water depths and flow velocities were constructed at Miyanaka Intake Dam. The width details are shown in Figure 2c,d.

| Fishway Type | | Gap (m) | Water Depth (m) | Width (m) | Velocity (m/s) | Discharge (m³/s) |
|---|---|---|---|---|---|---|
| Ice-harbor | General part | 0.25 | 0.24 | 8.0 | 1.27–1.69 | 1.637 |
| | Notch part | | 0.39 | | 1.57–2.43 | |
| Stair-type | – | 0.25 | 0.13 | 1.5 | 0.87–1.05 | 0.133 |
| Rock-ramp | Up to 2014 | 0.15 | 0.08 | 2.0 | 0.33 | 0.022 |
| | From 2015 | | 0.15 | | 0.64 | 0.071 |

Next, it was confirmed by one-way ANOVA in SPSS (Statistical Package for the Social Sciences) whether environmental factors had a great influence on the survey results or not among captured fish groups of *P. altivelis* and *T. hakonensis*. The numbers of captured *P. altivelis* and *T. hakonensis* were divided into three groups depending on frequency distribution in SPSS (Low = 1, Medium = 2, High = 3) (Table 4). The underlined data are a particularly important result. In the case of *P. altivelis*, a significant correlation is observed between captured fishes and water temperature in both groups, whereas SS and discharge do not show any significant correlation. In *T. hakonensis*, a significant correlation is found in captured fishes with SS for both the groups and discharge from low to high group. There is no significant correlation found with water temperature. These findings imply that environmental factors did not significantly affect the survey results.

**Table 4.** The results of multiple comparisons (Bonferroni) between the number of caught *P. altivelis* (a) and *T. hakonenis* (b) and environmental factors (discharge, ss, and temperature) are shown. The underlined results are important values indicating that environmental factors did not significantly affect the survey results.

| | | | | | | 95% Confidence Interval | |
|---|---|---|---|---|---|---|---|
| **(a)** | | | | | | | |
| Dependent Variable | (I) *P. altivelis* | (J) *P. altivelis* | Mean Difference (I–J) | Std. Error | Sig. | Lower Bound | Upper Bound |
| Discharge | 1.00 | 2.00 | −9.26 | 16.32 | 1.00 | −49.32 | 30.81 |
| | | 3.00 | 16.31 | 15.92 | 0.93 | −22.76 | 55.38 |
| | 2.00 | 1.00 | 9.26 | 16.32 | 1.00 | −30.81 | 49.32 |
| | | 3.00 | 25.57 | 16.93 | 0.41 | −15.99 | 67.12 |
| | 3.00 | 1.00 | −16.31 | 15.92 | 0.93 | −55.38 | 22.76 |
| | | 2.00 | −25.57 | 16.93 | 0.41 | −67.12 | 15.99 |
| SS | 1.00 | 2.00 | −14.67 | 9.83 | 0.42 | −38.78 | 9.45 |
| | | 3.00 | 3.47 | 9.58 | 1.00 | −20.05 | 27.00 |
| | 2.00 | 1.00 | 14.67 | 9.83 | 0.42 | −9.45 | 38.78 |
| | | 3.00 | 18.14 | 10.19 | 0.24 | −6.88 | 43.16 |
| | 3.00 | 1.00 | −3.47 | 9.58 | 1.00 | −27.00 | 20.05 |
| | | 2.00 | −18.14 | 10.19 | 0.24 | −43.16 | 6.88 |

**Table 4.** *Cont.*

**(a)**

| Dependent Variable | (I) *P. altivelis* | (J) *P. altivelis* | Mean Difference (I–J) | Std. Error | Sig. | 95% Confidence Interval | |
|---|---|---|---|---|---|---|---|
| | | | | | | Lower Bound | Upper Bound |
| Temperature | 1.00 | 2.00 | −0.90 * | 0.31 | 0.02 | −1.67 | −0.14 |
| | | 3.00 | −1.32 * | 0.30 | <0.001 | −2.06 | −0.57 |
| | 2.00 | 1.00 | 0.90 * | 0.31 | 0.15 | 0.14 | 1.67 |
| | | 3.00 | −0.41 | 0.32 | 0.62 | −1.20 | 0.38 |
| | 3.00 | 1.00 | 1.32 * | 0.30 | <0.001 | 0.57 | 2.06 |
| | | 2.00 | 0.41 | 0.32 | 0.62 | −0.38 | 1.20 |

**(b)**

| Dependent Variable | (I) *T. hakonensis* | (J) *T. hakonensis* | Mean Difference (I–J) | Std. Error | Sig. | 95% Confidence Interval | |
|---|---|---|---|---|---|---|---|
| | | | | | | Lower Bound | Upper Bound |
| Discharge | 1.00 | 2.00 | 29.85 | 12.48 | 0.06 | −0.56 | 60.27 |
| | | 3.00 | 37.35 * | 11.28 | 0.00 | 9.86 | 64.83 |
| | 2.00 | 1.00 | −29.85 | 12.48 | 0.06 | −60.27 | 0.56 |
| | | 3.00 | 7.50 | 13.21 | 1.00 | −24.69 | 39.68 |
| | 3.00 | 1.00 | −37.35 * | 11.28 | 0.00 | −64.83 | −9.86 |
| | | 2.00 | −7.50 | 13.21 | 1.00 | −39.68 | 24.69 |
| SS | 1.00 | 2.00 | 22.30 * | 7.16 | 0.01 | 4.86 | 39.74 |
| | | 3.00 | 23.79 * | 6.47 | 0.00 | 8.03 | 39.55 |
| | 2.00 | 1.00 | −22.30 * | 7.57 | 0.01 | −39.74 | −4.86 |
| | | 3.00 | 1.49 | 6.47 | 1.00 | −16.96 | 19.95 |
| | 3.00 | 1.00 | −23.79 * | 6.47 | 0.00 | −39.55 | −8.03 |
| | | 2.00 | −1.49 | 7.75 | 1.00 | −19.95 | 16.96 |
| Temperature | 1.00 | 2.00 | −0.28 | 0.34 | 1.00 | −1.10 | 0.55 |
| | | 3.00 | −0.24 | 0.31 | 1.00 | −0.98 | 0.51 |
| | 2.00 | 1.00 | 0.28 | 0.34 | 1.00 | −0.55 | 1.10 |
| | | 3.00 | 0.04 | 0.36 | 1.00 | −0.84 | 0.91 |
| | 3.00 | 1.00 | 0.24 | 0.31 | 1.00 | −0.51 | 0.98 |
| | | 2.00 | −0.04 | 0.36 | 1.00 | −0.91 | 0.84 |

* The mean difference is significant at the 0.05 level.

### 3.2. Utilization of Each Fishway

The numbers of captured individuals were averaged over four years (2012–2015). A total of 32 species and 11 species of bottom-dwelling fish were captured (related Table 5). As the survey period was selected based on the run-up period of *P. altivelis*, their captured numbers are the highest, and *P. altivelis*, *T. hakonensis*, and *O. platypus* are the dominant species. Table 5 confirms the tendency to catch bottom-dwelling fish and other fish species. An average of 75 individuals (approximately 83.0%) of bottom-dwelling fish (no. 22–32) selected the rock-ramp fishway. In particular, *Lethenteron sp.*, *Misgurnus anguillicaudatus*, *Misgurnus dabryanus*, *Cobitis biwae*, and *Liobagrus reini* were captured only on the rock-ramp fishway. For fish other than bottom-dwelling fish, an average of 3507 individuals (approximately 77.7%) chose the ice-harbor fishway, and an average of 987 individuals (approximately 21.9%) chose the stair-type fishway. Nine of the twenty-one species did not select the rock-ramp fishway, including *Carassius auratus langsdorfii*, *Hemibarbus barbus*, and *O. mykiss*.

**Table 5.** Numbers of fish caught in S-1 of each fishway during the period of four years from 2012 to 2015. The numbers shown are annual averages, and 1–21 are not bottom-dwelling fish.

| | Scientific Name | Ice-Harbor | Stair-Type | Rock-Ramp | Total |
|---|---|---|---|---|---|
| 1 | *Carassius auratus langsdorfii* | 0.5 | 0.3 | 0.0 | 0.8 |
| 2 | *Carassius* | 0.0 | 0.0 | 0.5 | 0.5 |
| 3 | *Opsariichthys platypus* | 149.8 | 163.5 | 4.0 | 317.3 |
| 4 | *Nipponocypris temminckii* | 0.3 | 2.0 | 0.0 | 2.3 |
| 5 | *Rhynchocypris logowskii steindachneri* | 0.8 | 11.5 | 4.8 | 17.0 |
| 6 | *Tribolodon nakamurai* | 0.5 | 0.0 | 0.0 | 0.5 |
| 7 | *Tribolodon hakonensis* | 455.8 | 59.5 | 1.3 | 516.5 |
| 8 | *Pseudourasbora parva* | 0.0 | 0.0 | 1.3 | 1.3 |
| 9 | *Sarcocheilichthys variegatus microoculus* | 3.5 | 0.3 | 0.0 | 3.8 |
| 10 | *Gnathopogon elongatus* | 0.0 | 0.8 | 2.3 | 3.0 |
| 11 | *Pseudogobio esocinus* | 8.8 | 18.8 | 0.0 | 27.5 |
| 12 | *Hemibarbus barbus* | 10.3 | 0.5 | 0.0 | 10.8 |
| 13 | *Squalidus chankaensis biwae* | 7.5 | 99.5 | 0.0 | 107.0 |
| 14 | *Cyprinidae* | 1.0 | 1.3 | 0.0 | 2.3 |
| 15 | *Plecoglossus altivelis* | 2826.8 | 623.0 | 0.5 | 3450.3 |
| 16 | *Salmo trutta* | 2.3 | 0.0 | 0.3 | 2.5 |
| 17 | *Salvelinus leucomaenis pluvius* | 3.5 | 0.3 | 0.3 | 4.0 |
| 18 | *Oncorhynchus mykiss* | 2.8 | 0.3 | 0.0 | 3.0 |
| 19 | *Oncorhynchus masou* | 30.3 | 5.0 | 1.8 | 37.0 |
| 20 | *Micropterus salmoides* | 0.0 | 0.0 | 0.8 | 0.8 |
| 21 | *Micropterus dolomieu* | 3.0 | 0.8 | 0.3 | 4.0 |
| | Subtotal | 3507.0 | 987.0 | 17.8 | 4511.8 |
| 22 | *Lethenteron* sp. | 0.0 | 0.0 | 0.3 | 0.3 |
| 23 | *Anguilla* | 0.0 | 0.3 | 0.0 | 0.3 |
| 24 | *Misgurnus anguillicaudatus* | 0.0 | 0.0 | 0.3 | 0.3 |
| 25 | *Misgurnus dabryanus* | 0.0 | 0.0 | 0.3 | 0.3 |
| 26 | *Misgurnus* | 0.0 | 0.3 | 0.0 | 0.3 |
| 27 | *Cobitis biwae* | 0.0 | 0.0 | 2.8 | 2.8 |
| 28 | *Pelteobagrus nudiceps* | 1.0 | 1.3 | 0.0 | 2.3 |
| 29 | *Silurus asotus* | 0.3 | 0.0 | 0.0 | 0.3 |
| 30 | *Liobagrus reini* | 0.0 | 0.0 | 2.5 | 2.5 |
| 31 | *Cottus pollux* | 0.5 | 9.3 | 22.5 | 32.3 |
| 32 | *Rhinogobius kurodai* | 0.3 | 2.5 | 47.0 | 49.8 |
| | Subtotal | 2.0 | 13.5 | 75.5 | 91.0 |
| | Total | 3509 | 1001 | 93 | 4603 |

Figure 4 depicts the ratio of the number of catches for each fishway. Except for bottom-dwelling fish, the ratio is shown only for fish in which 10 or more individuals were caught on average annually. Most fish other than bottom-dwelling fish used the ice-harbor and stair-type fishways. *Rhynchocypris logowskii steindachneri*, which mostly utilized the rock-ramp fishway, is a small fish. Most bottom-dwelling fish used the rock-ramp fishway. *Pelteobagrus nudiceps* and *S. asotus*, which mostly utilized the ice-harbor fishway, are large

bottom-dwelling fish. The relatively gentle stair-type fishway was also used by bottom-dwelling fish such as *Anguilla* sp., *Misgurnus* sp., and *C. pollux*. Thus, fish with relatively strong swimming abilities tended to use the ice-harbor fishway easily without employing the rock-ramp fishway. Further, bottom-dwelling fish with relatively low swimming abilities utilized the slow-flowing rock-ramp fishway instead of the ice-harbor fishway.

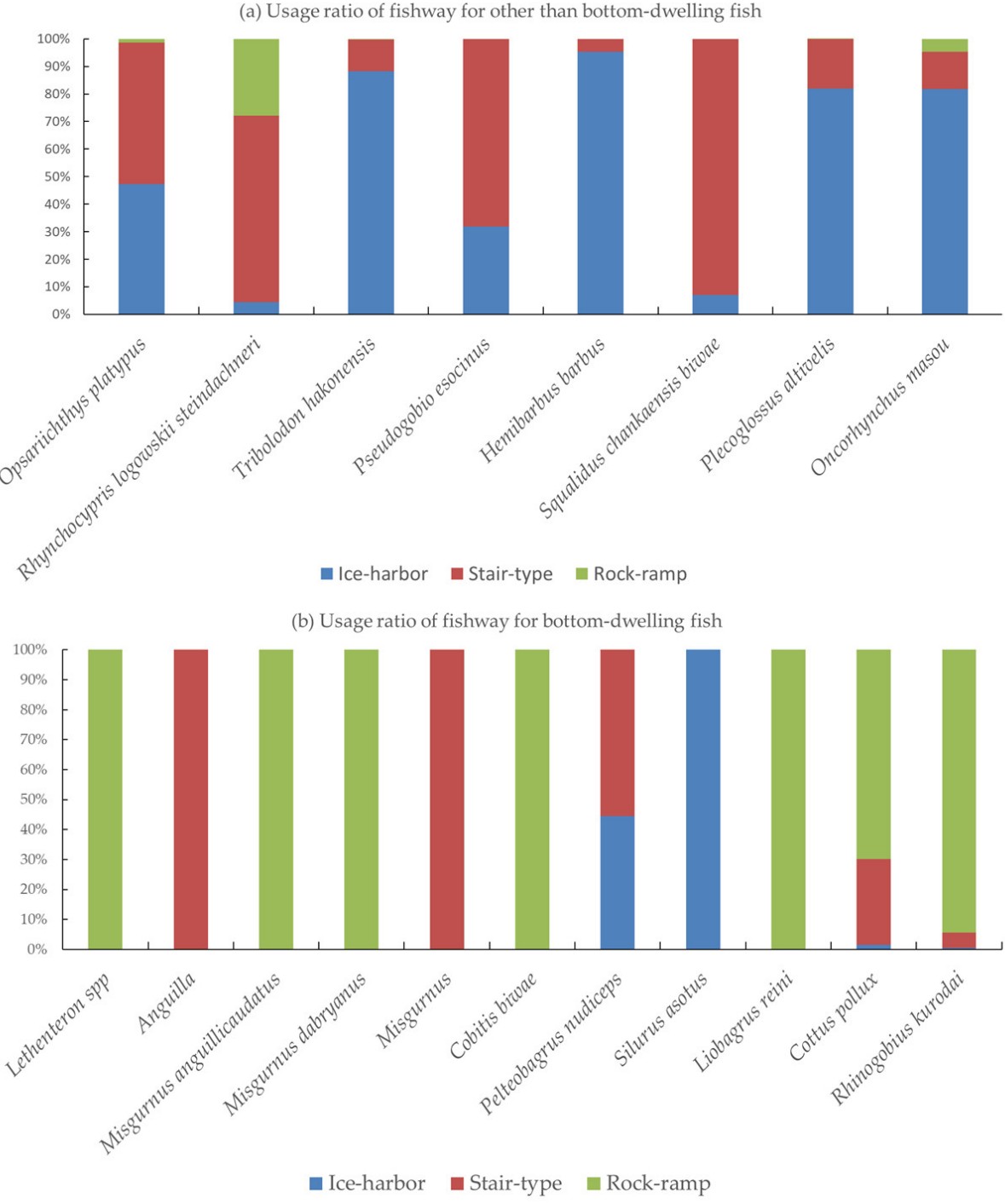

**Figure 4.** Usage ratio of the fishway for each caught fish over four years (2012–2015): (**a**) usage ratio for other than bottom-dwelling fish; (**b**) usage ratio for bottom-dwelling fish.

### 3.3. Habitat of Bottom-Dwelling Species

For frequent fishway users, such as *P. altivelis*, *T. nakamurai*, and *O. platypus*, many individuals were found in the ice-harbor fishway, but the fishway was used only for passage.

Some bottom-dwellers consistently inhabited the rock-ramp fishway. The numbers of captured individuals at the midpoints of the rock-ramp fishway are the averages over these four years (2012 to 2015). A total of 23 species were confirmed. According to the results of S-1, S-2, and S-3, the species that were not captured by S-1, such as *H, barbus* and *Cyprinidae sp.*, were captured by S-2 and S-3. For *T. hakonensis*, *S. leucomaenis pluvius*, *M. anguillicaudatus*, *M. dabryanus*, and *C. biwae*, the S-2 and S-3 numbers are higher than the S-1 number (related Table 6). This finding suggests that these species and their fry utilize the rock-ramp fishway as a habitat as well as for migration.

**Table 6.** Average numbers of catches at the midpoints of the rock-ramp fishway from 2012 to 2015. Fish 1–15 are not bottom-dwelling fish. Column S-1 presents the results from the upstream end, and the data overlap with the results in Table 4. Columns S-2 and S-3 depict the results from the midpoints of the rock-ramp fishway. Column S-4 lists the results from a point near the downstream submerged part at the entrance of the fishway and includes fish that did not run up the rock-ramp fishway.

| | Scientific Name | Average (2012–2015) | | | | |
|---|---|---|---|---|---|---|
| | | S-1 | S-2 | S-3 | S-4 | Total (S-2 and 3) |
| 1 | *Carassius* sp. | 0.5 | 0.3 | 0.0 | 0.3 | 0.3 |
| 2 | *Opsariichthys platypus* | 4.0 | 0.5 | 0.5 | 8.5 | 1.0 |
| 3 | *Rhynchocypris logowskii steindachneri* | 4.8 | 0.0 | 0.3 | 0.3 | 0.3 |
| 4 | *Tribolodon hakonensis* | 1.3 | 1.8 | 5.3 | 93.3 | 7.0 |
| 5 | *Pseudorasbora parva* | 1.3 | 0.3 | 0.0 | 0.0 | 0.3 |
| 6 | *Gnathopogon elongatus* | 2.3 | 0.3 | 0.0 | 0.0 | 0.3 |
| 7 | *Pseudogobio esocinus* | 0.0 | 0.0 | 0.0 | 0.3 | 0.0 |
| 8 | *Hemibarbus barbus* | 0.0 | 4.0 | 7.0 | 11.8 | 11.0 |
| 9 | *Cyprinidae* sp. | 0.0 | 14.3 | 15.3 | 66.8 | 29.5 |
| 10 | *Plecoglossus altivelis* | 0.5 | 0.0 | 0.0 | 0.0 | 0.0 |
| 11 | *Salmo trutta* | 0.3 | 0.0 | 0.0 | 0.5 | 0.0 |
| 12 | *Salvelinus leucomaenis pluvius* | 0.3 | 0.3 | 0.3 | 0.0 | 0.5 |
| 13 | *Oncorhynchus masou* | 1.8 | 0.3 | 0.8 | 0.0 | 1.0 |
| 14 | *Micropterus salmoides* | 0.8 | 0.0 | 0.0 | 0.0 | 0.0 |
| 15 | *Micropterus dolomieu* | 0.3 | 0.0 | 0.3 | 4.3 | 0.3 |
| 16 | *Lethenteron* sp. | 0.3 | 0.0 | 0.0 | 0.0 | 0.0 |
| 17 | *Misgurnus anguillicaudatus* | 0.3 | 0.0 | 0.5 | 3.0 | 0.5 |
| 18 | *Misgurnus dabryanus* | 0.3 | 1.0 | 2.8 | 5.8 | 3.8 |
| 19 | *Misgurnus* sp. | 0.0 | 0.0 | 0.0 | 0.3 | 0.0 |
| 20 | *Cobitis biwae* | 2.8 | 0.5 | 4.3 | 1.5 | 4.8 |
| 21 | *Liobagrus reini* | 2.5 | 0.0 | 0.0 | 0.5 | 0.0 |
| 22 | *Cottus pollux* | 22.5 | 0.5 | 2.0 | 10.0 | 2.5 |
| 23 | *Rhinogobius kurodai* | 47.0 | 0.3 | 0.0 | 14.0 | 0.3 |
| | total | 93 | 24 | 39 | 221 | 63 |

The survey at the points upstream and downstream from Miyanaka Intake Dam was conducted every spring (end of June), summer (end of August), and autumn (end of October) from 2009 to 2015. With the structural improvement of the fishway in 2012, a rock-ramp fishway was newly installed, and ice-harbor and stair-type fishways were constructed. In total, 31 species other than bottom-dwelling fish were captured, and 12 species of bottom-dwelling fish were captured (related Table 7). *H. barbus*, *O. platypus*, and *T. hakonensis* dominated both upstream and downstream. The numbers of species other than bottom-dwelling fish confirmed at the upstream survey point were 26 from 2009 to 2011 and 25 from 2012 to 2015. The numbers of bottom-dwelling fish confirmed at the upstream survey point were eight from 2009 to 2011 and nine from 2012 to 2015. The numbers of species other than bottom-dwelling fish confirmed at the downstream survey point were 16 from 2009 to 2011 and 20 from 2012 to 2015. Ten species of bottom-dwelling fish were confirmed at the downstream survey points from 2009 to 2011, and nine species from 2012 to 2015. The numbers of fish species caught upstream and downstream were compared. From 2009 to 2011 (before fishway improvement), 23 of the 43 species

(approximately 54%) were caught both upstream and downstream. From 2012 to 2015 (after fishway improvement), 27 of the 43 species (approximately 63%) were caught both upstream and downstream. Population interactions at these sites, upstream and downstream from the dam, are important for maintaining genetic diversity. Except for bottom-dwelling fish, the populations of 13 of the 29 species increased at the upstream point (approximately 45%). At the downstream point, the populations of 14 out of 22 species increased (approximately 64%). The populations of six out of ten species of the bottom-dwelling fish increased (approximately 60%) at the upstream point. At the downstream point, the populations of seven out of eleven species increased (approximately 64%).

**Table 7.** Survey results from 2009 to 2015 at the upstream and downstream points of Miyanaka Intake Dam. The numbers of captured individuals show the average values for each period. These results were obtained by conducting two-day surveys in June, August, and October of each year.

| Scientific Name | Upstream | | Downstream | |
|---|---|---|---|---|
| | **Average Individuals (2009 to 2011)** | **Average Individuals (2012 to 2015)** | **Average Individuals (2009 to 2011)** | **Average Individuals (2012 to 2015)** |
| *Carassius auratus buergeri* | 0.3 | 0.0 | 0.0 | 0.8 |
| *Carassius auratus langsdorfii* | 17.7 | 8.3 | 1.3 | 1.5 |
| *Carassius cuvieri* | 0.3 | 1.3 | 0.0 | 0.3 |
| *Carassius* sp. | 3.0 | 5.8 | 0.0 | 0.3 |
| *Cyprinus carpio* | 3.7 | 7.0 | 0.3 | 7.8 |
| *Gnathopogon elongatus* | 42.7 | 10.3 | 2.3 | 5.5 |
| *Hemibarbus barbus* | 205.0 | 388.8 | 39.0 | 153.5 |
| *Nipponocypris temminckii* | 24.3 | 20.8 | 0.0 | 0.0 |
| *Opsariichthys platypus* | 506.3 | 213.8 | 633.3 | 103.0 |
| *Opsariichthys* sp. | 6.0 | 0.0 | 0.0 | 0.0 |
| *Pseudogobio esocinus* | 9.0 | 10.3 | 15.3 | 10.0 |
| *Pseudorasbora parva* | 61.7 | 40.8 | 20.3 | 14.0 |
| *Rhodeus ocellatus ocellatus* | 6.0 | 0.3 | 2.0 | 0.3 |
| *Rhynchocypris logowskii steindachneri* | 13.0 | 22.8 | 3.3 | 6.8 |
| *Sarcocheilichthys variegatus microoculus* | 0.3 | 0.0 | 0.0 | 0.0 |
| *Squalidus chankaensis biwae* | 2.0 | 1.8 | 13.7 | 0.3 |
| *Squalidus* sp. | 5.7 | 15.0 | 0.0 | 0.0 |
| *Cyprinidae* sp. | 35.7 | 0.8 | 0.0 | 0.3 |
| *Tribolodon nakamurai* | 1.3 | 1.0 | 0.3 | 0.8 |
| *Tribolodon hakonensis* | 185.0 | 527.5 | 69.0 | 88.3 |
| *Tribolodon* sp. | 2.3 | 0.0 | 0.0 | 0.0 |
| *Plecoglossus altivelis* | 9.3 | 10.8 | 13.3 | 5.5 |
| *Lepomis macrochirus* | 0.0 | 0.5 | 0.0 | 0.0 |
| *Micropterus dolomieu* | 12.0 | 28.8 | 7.3 | 19.3 |
| *Micropterus salmoides* | 1.0 | 0.3 | 0.0 | 0.0 |

**Table 7.** *Cont.*

| Scientific Name | Upstream | | Downstream | |
|---|---|---|---|---|
| | Average Individuals (2009 to 2011) | Average Individuals (2012 to 2015) | Average Individuals (2009 to 2011) | Average Individuals (2012 to 2015) |
| *Channa argus* | 0.0 | 0.0 | 0.3 | 0.0 |
| *Oncorhynchus keta* | 0.0 | 1.0 | 0.0 | 0.5 |
| *Oncorhynchus masou* | 2.7 | 0.5 | 0.3 | 0.0 |
| *Oncorhynchus mykiss* | 0.3 | 0.3 | 0.0 | 0.3 |
| *Salmo trutta* | 0.0 | 0.5 | 0.0 | 0.0 |
| *Salvelinus* sp. | 0.0 | 0.0 | 0.0 | 0.0 |
| *Anguilliformes* sp. | 0.0 | 0.0 | 0.3 | 0.0 |
| *Liobagrus reini* | 0.3 | 0.8 | 3.3 | 8.3 |
| *Cobitis biwae* | 1.0 | 4.8 | 15.7 | 38.5 |
| *Lefua echigonia* | 0.0 | 0.3 | 0.0 | 0.0 |
| *Misgurnus anguillicaudatus* | 13.7 | 5.8 | 7.0 | 5.3 |
| *Paramisgurnus dabryanus* | 0.0 | 1.3 | 0.0 | 1.8 |
| *Rhinogobius fluviatilis* | 0.0 | 0.0 | 0.3 | 0.0 |
| *Rhinogobius kurodai* | 6.7 | 1.3 | 1.7 | 2.0 |
| *Lethenteron* spp | 0.7 | 0.0 | 4.0 | 1.0 |
| *Cottus pollux* | 2.7 | 7.0 | 1.0 | 4.3 |
| *Pelteobagrus nudiceps* | 1.3 | 1.5 | 2.3 | 3.5 |
| *Siluriformes* sp. | 7.0 | 2.8 | 6.3 | 7.3 |

*3.4. Effects of the Rock-Ramp Fishway*

To confirm the effects of the rock-ramp fishway, the catch results for the ice-harbor and stair-type fishways were combined and compared with those of the rock-ramp fishway. The capture survey results for each fishway during the first four years (2012–2015) after the construction of the new rock-ramp fishway are shown (Table 8). Among the 19 species of swimming fish, *P. parva* (t (1.981), df = 116, $p = 0.025 < 0.05$) and *M. salmoides* (t (1.981), df = 116, $p = 0.083 < 0.1$) were predominantly caught in the rock-ramp fishway. Among the other 17 species, the numbers caught on the rock-ramp fishway were not significantly different from those on the other fishways. All *P. parva* and *M. salmoides* were captured on the rock-ramp fishway. All of these were immature fish with body lengths of 61 mm or less. Although not significant, *G. elongatus* (t (1.974), df = 168, $p = 0.128 > 0.1$), which was often caught in the rock-ramp fishway, was also an immature fish with a body length of 61 mm or less. Among the bottom-dwelling fish, four of the ten species—*C. biwae* (t (1.981), df = 116, $p = 0.002 < 0.05$), *L. reini* (t (1.981), df = 116, $p = 0.077 < 0.1$), *C. pollux* (t (1.978), df = 136, $p = 0.040 < 0.05$) and *R. kurodai* (t (1.980), df = 120, $p = 2.12 \times 10^{-12} < 0.05$)—were caught more significantly in the rock-ramp fishway. *Pelteobagrus nudiceps*, a relatively large bottom-dwelling fish, was always captured on the ice-harbor fishway (t (1.981), df = 116, $p = 0.002 < 0.05$). It is considered that this situation occurred because the fish were long and had strong swimming abilities. Five species, *Lethenteron* sp., *Anguillidae*, *M. anguillicaudatus*, *M. dabryanus*, and *S. asotus*, were not significantly evaluated because only one individual was captured in the four survey years.

**Table 8.** Results of capture surveys in each fishway for four years (2012–2015) after the new rock-ramp fishway was established. The number of captured individuals shows the average value for 4 years. (Ice-harbor and stair-type fishways: I + S, rock-ramp fishway: R).

| Scientific Name | Fishway Type | Average Number of Individuals (Population) | d.f. | t | P |
|---|---|---|---|---|---|
| *Carassius auratus langsdorfii* | I + S | 3 | 116 | 1.981 | 0.083 |
| | R | 0 | | | |
| *Carassius* sp. | I + S | 0 | 116 | 1.981 | 0.158 |
| | R | 2 | | | |
| *Opsariichthys platypus* | I + S | 1253 | 116 | 1.981 | 0.000 |
| | R | 16 | | | |
| *Nipponocypris temminckii* | I + S | 9 | 116 | 1.981 | 0.012 |
| | R | 0 | | | |
| *Rhynchocypris logowskii steindachneri* | I + S | 49 | 157 | 1.975 | 0.028 |
| | R | 19 | | | |
| *Tribolodon nakamurai* | I + S | 2 | 116 | 1.981 | 0.158 |
| | R | 0 | | | |
| *Tribolodon hakonensis* | I + S | 2061 | 116 | 1.981 | 0.000 |
| | R | 5 | | | |
| *Pseudorasbora parva* | I + S | 0 | 116 | 1.981 | 0.025 |
| | R | 5 | | | |
| *Sarcocheilichthys variegatus microoculus* | I + S | 15 | 116 | 1.981 | 0.000 |
| | R | 0 | | | |
| *Gnathopogon elongatus* | I + S | 3 | 168 | 1.974 | 0.128 |
| | R | 9 | | | |
| *Pseudogobio esocinus* | I + S | 110 | 116 | 1.981 | 0.000 |
| | R | 0 | | | |
| *Hemibarbus barbus* | I + S | 43 | 116 | 1.981 | 0.000 |
| | R | 0 | | | |
| *Squalidus chankaensis biwae* | I + S | 428 | 116 | 1.981 | 0.009 |
| | R | 0 | | | |
| *Cyprinidae* sp. | I + S | 9 | 116 | 1.981 | 0.072 |
| | R | 0 | | | |
| *Plecoglossus altivelis* | I + S | 13,799 | 116 | 1.981 | 0.001 |
| | R | 2 | | | |
| *Salmo trutta* | I + S | 9 | 138 | 1.977 | 0.019 |
| | R | 1 | | | |
| *Salvelinus leucomaenis pluvius* | I + S | 15 | 128 | 1.979 | 0.002 |
| | R | 1 | | | |
| *Oncorhynchus mykiss* | I + S | 12 | 116 | 1.981 | 0.028 |
| | R | 0 | | | |

**Table 8.** *Cont.*

| Scientific Name | Fishway Type | Average Number of Individuals (Population) | d.f. | t | P |
|---|---|---|---|---|---|
| *Oncorhynchus masou* | I + S | 141 | 124 | 1.979 | 0.000 |
| | R | 7 | | | |
| *Lethenteron* sp. | I + S | 1 | 116 | 1.981 | 0.319 |
| | R | 0 | | | |
| *Anguilla* sp. | I + S | 1 | 116 | 1.981 | 0.319 |
| | R | 0 | | | |
| *Misgurnus anguillicaudatus* | I + S | 0 | 116 | 1.981 | 0.319 |
| | R | 1 | | | |
| *Misgurnus dabryanus* | I + S | 0 | 116 | 1.981 | 0.319 |
| | R | 1 | | | |
| *Misgurnus* sp. | I + S | 1 | 116 | 1.981 | 0.319 |
| | R | 0 | | | |
| *Cobitis biwae* | I + S | 0 | 116 | 1.981 | 0.002 |
| | R | 11 | | | |
| *Pelteobagrus nudiceps* | I + S | 9 | 116 | 1.981 | 0.002 |
| | R | 0 | | | |
| *Silurus asotus* | I + S | 1 | 116 | 1.981 | 0.319 |
| | R | 0 | | | |
| *Liobagrus reini* | I + S | 0 | 116 | 1.981 | 0.077 |
| | R | 10 | | | |
| *Cottus pollux* | I + S | 39 | 136 | 1.978 | 0.040 |
| | R | 90 | | | |
| *Rhinogobius kurodai* | I + S | 10 | 120 | 1.980 | 0.000 |
| | R | 189 | | | |

## 4. Discussion

### 4.1. Fishway Types for Target Fish Species

Regarding the sustainability of fishery resources, usually, only commercially valuable fish species are considered conservation targets. Salmonid fish, such as *Salmo salar*, *S. trutta*, *O. tshawytscha*, and *O. mykiss* are typically targeted in North America and Europe [13,24,25,76,77]. Because the fish species targeted in both the US and Japan are relatively capable swimmers, the present ice-harbor and stair-type fishways are sufficient. For these species, conduit-type fishways, which involve relatively high flows regardless of their design specifics, are often selected and installed at the sides of dams [7,26,27,78–82].

The major species found in the fishways were *P. altivelis*, *T. hakonensis*, and *O. platypus*. The former two species were in the process of diadromous migration, whereas *O. platypus* is purely a freshwater species. The large numbers of *O. platypus* are likely the result of a recent population increase owing to river channel rehabilitation. Among the major users of the ice-harbor and stair-type fishways, *P. altivelis* and *T. hakonensis* mainly employed the ice-harbor fishway, which had the highest flow velocity. Meanwhile, *O. platypus* used the stair-type fishway more than the ice-harbor fishway. Because the sizes of the individuals vary slightly between these species, these differences in preference are likely associated with their swimming abilities. Our results indicate that flow velocity, regardless of fishway type, is more important than water depth and flow rate for many of the target fish species [83].

*4.2. Effects on Benthic Species and Species with Low Swimming Abilities*

Large conduit-type fishways are important for the migration of commercially important fish. However, large conduit-type fishways alone are not sufficient to protect the biodiversity of river ecosystems inhabited by benthic fish and species with low swimming abilities. In the Miyanaka Intake Dam, the rock-ramp fishway was constructed in parallel with the ice-harbor and stair-type fishways. The rock-ramp fishway was designed with a gentle slope and low flow velocity. To imitate a natural river, instead of creating a step with a concrete wall, boulders were laid down to form a slope [84]. Stillwater zones have emerged in the spaces behind the cobbled slope, and the loose main flow and quiet space help small swimming fish run up and bottom-dwelling fish inhabit the fishway.

The rock ramp fishway, with its continuous low walls, low water depth, and low flow velocity, also appears to be used as a habitat for bottom-dwelling species [85,86]. The habitats of bottom-dwelling species vary—gravelly, muddy, covered with submerged plants—in accordance with their lifestyles, and the bottom conditions are critical [87]. Our results support the steady existence of bottom dwellers *C. pollux* and *R. kurodai* in the rock-ramp fishway. They are freshwater fish that normally live between pieces of gravel/cobble. However, despite being bottom dwellers, *C. biwae* and *M. anguillicaudatus* were not found in the rock-ramp fishway because they prefer a fine sand bottom to a stony bottom and did not use the fishway as a habitat [88–91].

The entrances of the fishways were installed 1 m above the upstream bottom. Thus, even during floods, new sediments do not enter the fishways, except for silts, but they do not accumulate at the bottom owing to high velocity. Because the originally existing sand was gradually washed away and only the originally added cobbles remained, the rock-ramp fishway was not available to sand-bottom dwellers as a habitat. Overall, the rock-ramp fishway was more effective for gravel-/cobble-bed dwellers than for sand-bottom species. Other than bottom-dwelling species, species with weak swimming abilities and those that are unable to use the stair- or ice-harbor-type fishways require low-flow fishways. High-velocity zones affect feeding opportunities [92] and the amount of feeding that can be accomplished [93–95].

Small swimming species, such as *Gnathopogon elongatus*, *Pseudorasbora parva*, and *Squalidus chankaensis biwae* are freshwater species. Considering that they were caught in the rock-ramp fishway, they likely utilized this fishway for upstream and downstream movement, but they were not found to be regular dwellers. Hence, rock-ramp-type fishways are especially preferable for the movement of small fish and as habitats for bottom-dwelling fish [96–100]. In other words, rock-ramp fishways function as not only places of movement but also stable habitats.

*4.3. Effectiveness of Fishways*

Figure 5a–d present the numbers of individuals found in the upstream zone, inside the fishway, and in the downstream zone in different years. For the species for which more than 10 individuals were found, *C. pollux* (Figure 5a) and *R. kurodai* (Figure 5b) were mostly found in the rock-ramp fishway; no individuals were found in either the upstream or downstream zones. The confirmed numbers of individuals of both species have increased since 2012 when the fishway was improved. Meanwhile, *C. biwae* (Figure 5c) and *M. anguillicaudatus* (Figure 5d) were found mostly in the downstream zone (*C. biwae*) and were present in both the upstream and downstream zones but not in the fishways. Many *C. biwae* and *M. anguillicaudatus* were confirmed upstream and downstream from the dam. Although the number of individuals in the fishway increased slightly after 2012 when the fishway was improved, this number is smaller than the upstream and downstream numbers.

The rock-ramp fishway was newly installed to run up the bottom-dwelling fish. Consequently, it was adequate for increasing the numbers of inhabitants in the upstream and downstream areas. However, in reality, it was also used as a habitat for the bottom-dwelling fish and as a moving space for short fish. Therefore, it was confirmed that the rock-ramp fishway with low flow velocity substantially contributed to the run-up and habitat of bottom-dwelling fish and small fish with weak swimming abilities. Our results also indicate the importance of the rock-ramp fishway in this aspect.

When a fishway is installed, it is thought that setting the flow velocity and water depth according to the target fish species and a combination of multiple fishways would be effective. Our study, which focuses on the flow velocity of the fishway and explores the effects of the rock-ramp fishway, has shown useful results. As no previous report has proven the effectiveness of rock-ramp fishways and combining fishways, our findings may be useful for future fishway structural improvement plans for various fish, including demersal fish. Ensuring the mobility and habitation of diverse fish contributes to the harmony between power generation and the river environment. We propose utilizing our study results for facility managers of river crossing structures who aim to establish an environment where fish can live upstream and downstream of the dam without stress.

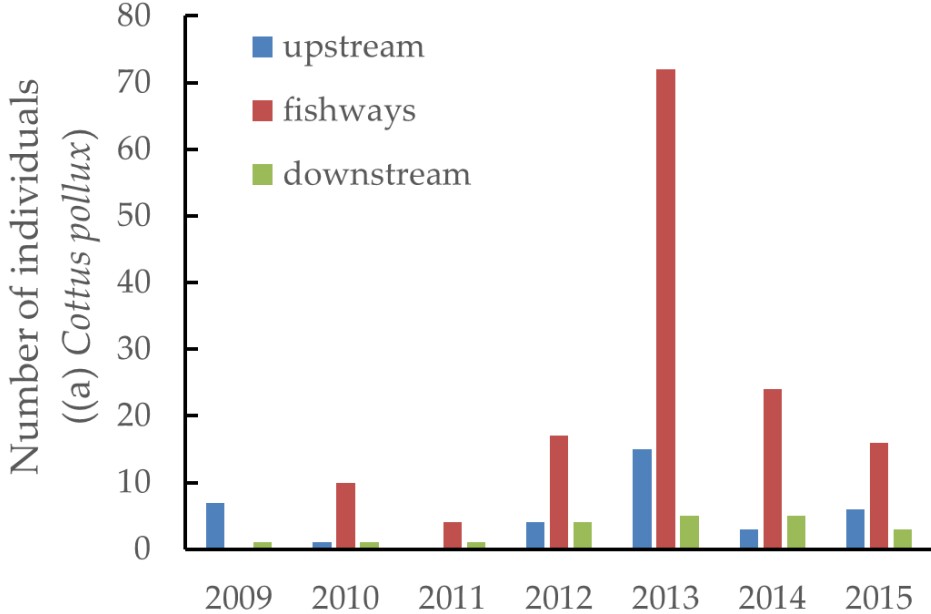

**Figure 5.** *Cont.*

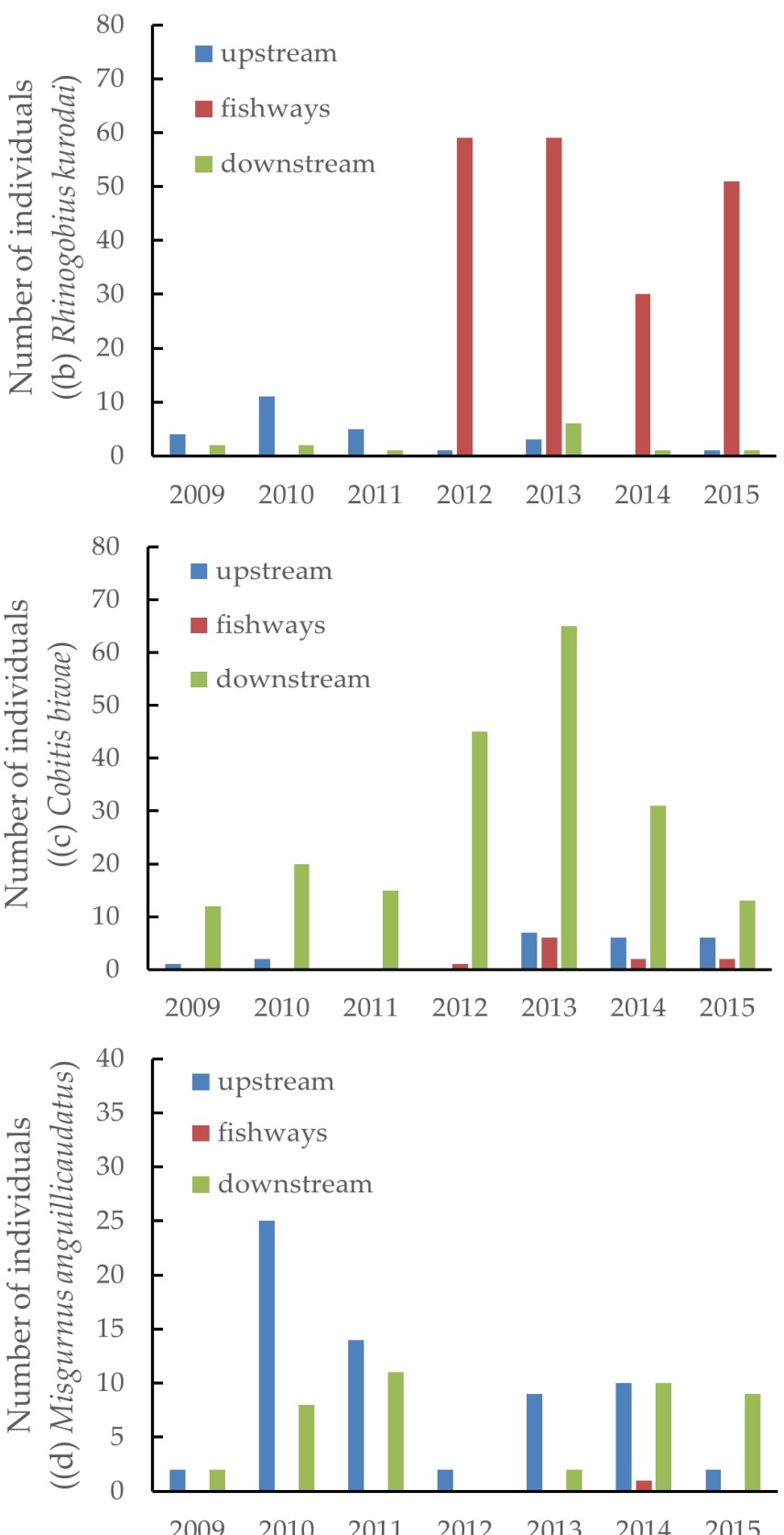

**Figure 5.** Numbers of bottom-dwelling species (**a**) *C. pollux* and (**b**) *R. kurodai* in the upstream zone, fishways, and downstream zone, as confirmed by the 2009–2015 survey. Independent numbers of bottom-dwelling species (**c**) *C. biwae* and (**d**) *M. anguillicaudatus* in the upstream zone, fishways, and downstream zone, as confirmed by the 2009–2015 survey.

## 5. Conclusions

The effects of the newly constructed rock-ramp fishway are investigated in this study. The new rock-ramp fishway is effective both as bottom-dwelling fish habitat and a moving environment for small fish with weak swimming abilities. It is foremost to set the flow velocity and water depth according to the target fish species for establishing a fish way.

Our study was established based on an understanding of the upstream and downstream of the dam and the habitat of fish in the fishway. However, we understand that there is room for further consideration and limitations from the perspective of understanding the habitats of fish upstream and downstream of the dam. Furthermore, we also understand that the survey methods for upstream and downstream of the dam and the fishways were different, so it was not possible to continuously evaluate the abundance of upstream, fishways, and downstream. While practicing adaptive management for 10 years, we also continue to understand the fish habitat upstream and downstream of the dam at slightly different points. Therefore, it is expected that better understanding can be achieved in the future.

**Author Contributions:** Conceptualization, T.M. and T.A.; methodology, T.M. and T.A.; software, T.M. and T.A.; validation, T.M. and T.A.; formal analysis, T.M. and T.A.; investigation, T.M., M.N. and T.A.; resources, T.M. and M.N.; data curation, T.M., T.A. and M.R.; writing—original draft preparation, T.M.; writing—reviewing and editing, T.M. and T.A.; visualization, T.M., T.A. and M.R.; supervision, T.M. and T.A.; project administration, M.N.; funding acquisition, T.M. and M.N. All authors have read and agreed to the published version of the manuscript.

**Funding:** This research received no external funding.

**Institutional Review Board Statement:** This survey has been approved annually by the Ministry of Land, Infrastructure, Transport and Tourism, Japan (notification of river use during the survey period and receipt) and Niigata Prefecture, Japan (application to catch fish and permission, example; Permit number, Special No.24).

**Informed Consent Statement:** Not applicable for studies not involving humans.

**Data Availability Statement:** The authors highly appreciate and state that data will be available for everyone upon request without undue reservation.

**Acknowledgments:** We would like to thank the Ministry of Land, Infrastructure, Transport and Tourism, Shinanogawa River Office for providing data on the ecology of fish, and the Miyanaka Intake Dam Fishway Structure Improvement Review Committee for advice on fish ecology.

**Conflicts of Interest:** The authors declare no conflict of interest.

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
