# Peer review of "Effectiveness of New Rock-Ramp Fishway at Miyanaka Intake Dam Compared with Existing Large and Small Stair-Type Fishways"

_water, doi:10.3390/w14131991_

Round 1

Reviewer 1 Report

I am really sorry to say that I still cannot recommend this paper for publication. The writing still mixes sections together, this time results and discussion. I think that overall, the paper has been improved by two rounds of reviewers, but I recommend seek of help of experienced scientist to make the formatting right.

Minor comments:

L154 – Change to “target water depth was achieved”

L156 and L158 – was improved -> was modified

Figure 2 – upper section – small letters, hard to read in print, mixing starting uppercase and lowercase

L296 – Delete “The survey method is shown in Table 1.” and reference Table in the text

L306 – Delete “The flow velocity measurement results are listed in Table 2.” and reference Table in the text (related text...(Table 2))

Figure 4 and interconnected analyses – this should not be solved by regressions, but advanced general linear models or GAM models

L344 – same comment as L296 and L306

L386-8 – Please change reference to table according to my previous comments

L395 – Same comment as to previous table references

L443-456 – This belongs to results, all the tests. I recommend to contact experienced scientist for help to finalize this paper for publication

L557-571 – Again, result section in discussion

Author Response

Thank you for your response.

I have created an answer to your peer review.

Reviewer 2 Report

The manuscript was somehow improved since the first version but it still does not achieve merits usual for an international paper. Some parts are not given in a very efficient and clear way, there is a large overlap between different chapters and the discussion is quite weak.

Specific comments:

Line 17 Use spectrum of methods instead of “ set nets and thigh nets”

Line 19 Migration instead of swimming

Line 21 Also instead of Thus

Line 498-57 awkward flow of ideas

Lines 62-66 belongs to Material and methods

Lines 77-81 delete

Line 88 the mention of slope 1/269 confusing and probably not relevant

Lines 90-94 needs more clear formulation

Line 101 water levels instead of areas?

Figure 1 Is the river flowing North or south? From (a) it looks flowing North, from (c) south.

Fishway designs this chapter still mixes several fishways 2- historical periods not in efficient way.

It would be good to use table similar to Table 2 with historical periods clearly distinguished and all important parameters of relevant fishways including target species given. There were several changes of operation and construction in time and they must be clearly defined.

Line 183 what is The Mio muscle?

Line 207-215 repetitive, delete

Lines 260-261 The types and numbers of fish caught were recorded  - not necessary, delete.

Figure 3 and corresponding text.  The construction of traps is not described sufficiently and the photos are too small to see. Not possible to follow the experiment or to have an idea how efficient were the traps.

L- 272-274. The fish were released upstream from the dam even when captured at S4 for example?

Table 1 would be useful but is very carelessly done. The gear and effort needs to be much more precisely described. Dimensions, mesh sizes, construction, what are set nets and thigh nets? Bait at longlines? Why were not used electrofishing which would be an obvious choice?

The river upstream seems much larger than downstream (Figure 1 c). How much discharge is taken away to run Shinkansen train? Does the change of river size matter for sampling results?

Lines 312-333, significance of result should be given here, not at the discussion.

Line 320 the formula looks wrong to me 0,9*20 = 18.  The line is much higher at Figure 4 a.

I guess, using linear models are clearly wrong! (b) and (c) looks pretty hyperbolic.

Table 3 legend, add at S1 site.

Line 387 midpoints

Line 339 and 408 species instead of types

Line 396 also 2009-2011

Lines 399-401 sentence “To confirm the effects of the rock-ramp fishway, the results from 2009 to 2011 and from 2012 to 2015 were separated” can be deleted.

Line 422 midpoints

Line 426 near the downstream submerged part

Lines 484-489 shorten to one sentence

Lines 441-452, 542-583 Results

  1. 459-461 repeating, unnecessary.

Discussion some bottom dwelling fish were not captured upstream/ downstream because of different methods (boulders were not moved to get interstitial fish.

  1. 540 I do not understand: applying the format

Conclusions can be shortened.

Lines 627-631 acknowledge the help of several reviewers who put a lot of effort to improve your manuscript.

Author Response

(The authors gave the same response as above.)

Round 2

Reviewer 2 Report

The manuscript has again improved a bit.  I must say that I am a bit tired of it as I am reading it for the third time. So I concentrated my attention mainly on material and methods, discussion and conclusions.

Material and methods should be described the way that any knowledgeable professional can repeat the work according to the description given in the manuscript. If you read the description of fish sampling, it is quite clear that no one can repeat it having the description provided by the authors. I understand that the spectrum of methods applied here is rather diverse but this is not the justification for careless description of tools and procedures.

Line 19 migration instead of swimming

Fig 1c. is probably not oriented in a normal way (North = up). Put it rght like all other maps. Where is actually the intake?  Make the arrow.

Chapter 232 I have feeling that authors use the words “trap” “cage” and “basket” not in fully consistent way. Good description of the trap should be done. I understand that cage is the rear part of the trap photographed on the Figure 3c while the front end consists of a fish guiding funnel with short wings. Not sure what is the basket. Also further descriptions are not clear. The dimmensions and individual parts of the trap would be best placed in the photograph.

The world temporary is used many times, this is not necessary, just state the period of traps operation (probably line 277)

Line 271 Not sure what depth 0,6 m means.

Line 275-276: Not sure what means all the fish in the basket were caught every day.  All the captured fish were removed every day?

Line 286: Not sure what the partition net is and how it functioned.

Figure 3 A. The legend is very long and awkward. A lot of details should be given in the text or dimensions directly at the photographs.

Line 304-305 somehow contradicts with Line 275-276: so were the fish removed hourly or daily?

Line 305 what means survey points. Does this mean that the series of six traps were installed there?

Line 307 introduces “small catch basket” in fact none is large, please describe the set-up properly!

It looks that rock ramp fish pass had different traps but there are not photographed so one can see the function and structure nor they are clearly described. Numbers of traps at different parts of fish ways should be given clearly.

It looks like in points S1 –S4 both traps and landing nets were used. Syi ti more clearly and show the trap of rock ramp fish pass clearly.

Line. 324 Each survey reach was 1 km long

Table 2 Use habitat instead of location. What means the wand?

The effort was 10 times at each habitat?

Stationary nets completely unsufficiently described also the effort is not clear – how many nights in what habitats. In fact at habitat columns, you should use the numbers describing the effort instead of the circles!

The same for the gillnets.  Actually the gillnets with the two mesh 15 and 90mm are very unusual. What means used together. Like trammel nets? Describe better!

Longline use bait instead of food.

Tables 7 and 8 it is not clear what are the units of numbers.

Lines 580-588 belong to discussion.

Line 611 Even if you do not know the reviewers you ought to say: Authors greatly appreciate the hlp of three(?) anonymous reviewers who greatly improved the manuscript. or something like this.

Author Response

Thank you for your guidance.

This manuscript is a resubmission of an earlier submission. The following is a list of the peer review reports and author responses from that submission.

Round 1

Reviewer 1 Report

The comments and suggestions raised in the previous review were not satisfactorily answered by the authors. However, there are still far too many in my opinion. As a consequence of this lack of brevity in presentation, the paper risks losing the reader’s attention of the key findings.

  • Furthermore, the figures are not in a consistent style, with different sizes (e.g., Figures 1, 4, 5, 6, 7), fonts and styles (e.g., Figures 1, 2, 3, 4…). Figures should be further consolidated where possible and reproduced in a consistent style of appropriate quality for publication.
  • This time, abstract and conclusion are the mix of background and results, need to remove those.
  • Sections of the paper are disjointed with grammar errors, including tense issues, missing words, missing punctuation, and awkwardly constructed sentences.

Specific comments:

Line 44, before inserting “technical” in here, it is better to introduce “technical” in lines 38-39 first.

Lines 83-91, A long sentence! Better to rewrite this one in several sentences.

Line 145, “rush speed”, better to change with the “burst speed”.

Lines 193-197, Definitely, these are the general statements about “rock-ramp” fishway! Better to move them to the Introduction section. Also, authors should read some papers on mean and turbulent flow characteristics of rock-ramp fishway before putting these statements.

Lines 223-231, not clear, need to rewrite.

Line 276, “SS” need to define.

Reviewer 2 Report

Major comments:

I am sad to report that the manuscript titled “Effectiveness of rock-ramp fishways versus other types” still needs to be heavily modified before it can be published, even after the revision that came to me for a second round. Second sentence I wrote in the previous review report is still valid, that is that authors present interesting topic, but fail in many standards of scientific writing. The manuscript should be read by English native speaker to correct all peculiar sentences found in the paper. And in my opinion, streamlining of the manuscript should be further improved besides minor edits that I present in Minor comments below. 

 In the current version, results have statistical analyses, which can be further improved (specific comments).

Conclusions – Please focus more on general conclusions and future directions, not just on repetitive statements

Minor comments:

L42 – with panels

L49-50 – why is ownership by railway company in any context relevant to fish migration? Please clarify or delete

L55 – replace move by migrate

L67-69 – Does not fit in context of methods

L83 – L91 – Sentence longer that 3 rows in not easy to read. This sentence spans over NINE rows...

L109-110 – It would be good to directly present also water speed, the comparison of discharge can be misleading

L145 – burst speed?

L170-171 – Not really understand the meaning

L173 – Occurrence of what?

L206-215 – Scheme of measurement would help

L218 – 0.5 cm precision?

L225-226 – Rephrase and choose whether you want to use this sentence here or on L228

L252 – Confirmed <- recorded

L259 - ...duplicated in records

L268-272 – Please rephrase this section

L273 – The relationship between....and discharge (Figure 3). Change all references to figures in this format

L283-287 – Correlations does not make sense in seasonal data, where you have usually peak in the middle. It needs some polynomial relationship.

L260–304 – Parts of this text belong to results (correlations and figures)

L325 – Change reference to figure as described above

Figure 4 – legend text - percentage

Figure 5 – Y axis label?

L360 – figure reference

Figure 7 and 8 – capitalize first letters in y and x axis

L407–409 – Try to connect with other paragraph

L468-470 – Results and in need of rephrasing

L471-475 -needs rephrasing, and potentially should be moved to methods, not sure what type refers to in these sentences

L477-483 – methods

L484-500 – results

L502-504 – methods

Table 1 and 2 – species names need italics

Reviewer 3 Report

This manuscript presents several years of short-termed observations of fish in three types of fishways. Authors used diverse sampling protocols: traps in fishways, simple sampling within the rock ramp fishway and sampling of downstream and upstream community. Unfortunately, this complicated design is not presented very clearly. There are a lot of technical details which authors try to explain in words but this is not an efficient way of conveying them. Technical drawings would be much better. The language of the paper is often very awkward and needs thorough editing by native speaker who understand scientific style. Authors very often mix up results with material and methods and discussion. As it is now the paper should not be released, it needs thorough rewriting.

The manuscript consists of green and black text. The meaning of these colour changes is not clear to me.

The authors define in fact three functional groups of fish. Bottom dwelling fish are clear but “swimming” fish are bit strange (all fish are swimming). This group could be defined as not bound to bottom while bottom dwelling fish could be called obligatory bottom dwelling fish. Paper needs a Table showing the fish groups. Can be done in Table 2.

Detailed comments (note when pasting to this web intrface, L. for line changed to numbers, then ignore the first number on the row. Following number means Line No.)

  1. 17-19 Two Sentences “The rock ramp…habitat” not necessary.
  2. 19-20. Change: Thus, rock-ramp fishways can also provide habitat for certain species, such as C. pollux. There is no clear information on breeding in the manuscript.
  3. 48 meaning downstream – upstream continuity?
  4. 49-50 note about the only dam of railway company probably not relevant.
  5. 50-52 At the same time, in consideration of fish biodiversity a new rock-ramp type was established to facilitate migration and provide habitat to the bottom-dwelling fish.
  6. 54-55. Before the improvement, only two fishways, the ice-harbor fishway and the stair-type fishway, were installed, so the conditions were not favourable for bottom-dwelling fish upstream migration.
  7. 59 replace also target by targeting
  8. 75 is 100 mm to 150 mm grain size or thickness of layer?
  9. 81 better word than elevation gap. Difference?
  10. 82 better word than upstream and downstream gap
  11. 82-83 Any reference to support the statement?
  12. 83-91 Divide extremely long sentence.
  13. 83-107 technical descriptions difficult to follow. Needs good technical top and side drawing with explanations. Photos on Fig. 1 are not clear enough.

Use table to show gaps, depths, widths, velocities, discharge rates of individual fishways.

I am not sure to what extent the description of the fishways before improvement is really relevant to the message of this paper.

  1. 122-123 Not clear what you mean.
  2. 124-126 I can see only one fishway.
  3. 127 boulders diameter 0,15 m are not gravel.

Figure 2 Dimensions in mm?

  1. L. 147-162 Use table for the measurements.

L 159 from Fig. 1D I feel that the width of fishway flow changes along.

  1. 164-200 would red better with good drawings.
  2. 206-215 Put measuring points to a figure.
  3. 223 what means “were set”?
  4. 223-231 Make table. How do these values relate to values at lines 147-160?
  5. 234 What type of temperature sensors?
  6. 240-243 What means “counted”?
  7. 245-246 Put the position of traps at technical drawing of fishways.
  8. 250 Provide photograph or good drawing of the traps.
  9. 251 during?
  10. 250-251 awkward sentence.
  11. 252 What means “confirmed”?
  12. 252-253 Benthic bottom dwelling fish hide under stones and can easily escape visual observations and landing nets. Electric sampling would be more relevant
  13. 254-257 I am confused, are these different traps than mentioned at lines 247-249.
  14. 260-271 Interesting things, not so nicely described but I have feeling that this information is not being used in the paper.
  15. 276. SS s suspended solids? How were they determined?
  16. 279-304. Are results, not Material and Methods.

¨Figure 3, 4, these relationships need not to be explained by simple linear model. Axis label are too small. How is daily capture population defined?

  1. 305-315 Sampling sites need to be shown on a map.

L m314-316 How were al these diverse methods and results put together. Explain standard protocol. Explain the effort.

  1. 317-321 should be moved to survey timings, somewhere around line 243.
  2. 324 strange title of a chapter

Figure 4 is actually counted two times!

  1. 339-40 Figure 4B use the same colours for all three histograms. Otherwise it suggests that Stair type fishpass has something to do with other species of Figure 4a.
  2. 343 Percentage of P. altivelis in all fishways (2012-2015). Altogether 30 species of fish…

Figure 5 What is aberae? Fonts are too small. Use species abbreviations. Explain Y axis better.

Figure 6 Explain Y axis better. What are average individuals?  Standardize results more rigorously. Are these results from traps or from in-fishway sampling by landing nets?

  1. 356 delete “other”

Figure 7 missing

Figure 8 define effort for sampling numbers.

  1. 393-398 results!

Table 1 subheadings “number”

Table 2 lacks clear column headings.

  1. 423-428 confusing

L 455 pass the gap???

  1. 463-467 results
  2. 472-476 Material and methods?
  3. 476 and further (496) at are fish types?
  4. 499-501 So it may be actually small fish size which decides the usage of rock ramp fishway.
  5. 526-534 awkward expressions.

Reviewer 4 Report

Manuscript Number:  Water 1492906

Title:  Effectiveness of Rock-ramp Fishways Over Other Types

Reviewer comments:

The manuscript is focused on the assessment of three fishways: ice-harbour, pool-type and a rocky ramp bypass. Authors present an interesting and necessary study centered in native fish species in a region where this kind of works is not usual. Both circumstances, in my opinion, make the manuscript doubly interesting for fish conservation. However, I got several comments related to the paper aims, applications and results that deeply concern me.

Authors want to test the effectiveness of three fishways, searching difference in fish species and size passage. I think this is the main aim, but I have deducted it once I have finished the revision. This should be clear at the end of the Introduction, and it is not (Lines 55-60). This section does not guide readers through the manuscript and references are weak (lot of forced self-citations and absence of important and worldwide references about fishways: Larinier, DVWK, Clay, etc. and fishway assessment: Castro-Santos, Noonan, Romao, etc). The definition “stairs” does not exits in any fishway design handbook (probably you refer to a stepped/technical fish ladder pool-type or pool-and-weir fishways).

Methodology is not clear, and it was not easy form me make me an idea about the assessed fishway size and types. I miss more information as EDV, longitudinal section (or 3D), and accurate design data to compare with other works (e.g., you say in Conclusions that depth in ice-harbour’s is 0.24-0.39 m, probably this is depth at the notches). I would like to know more info about trap and trapping method. I feel it as weak because you compare results from different trapping methods in each fishway. You also merge this section with Results (Lines 279-320).

Results section is disappeared, and Discussion are results poorly commented.

Dear authors, I like your work and love fishways, but I must be honest in my revision. I think this paper could become in a great manuscript, but it is not publishable in this format. My recommendation about the manuscript is to reject it. This is a first draft, and I am sure you could change this study to a scientific paper, but you have a lot of work to do.

Some recommendations:

I am not an English speaker, but you should check the language with a native speaker.

Think in a more attractive and real tittle e.g.: Effectiveness of three fishway types for native fish fauna in the Shinano River (Japan).

Read a classical fishway design handbook https://www.fao.org/publications/card/en/c/fc11949e-9dc5-59a1-bf10-9e9480a3fe7f/   and some papers about fishways assessment: https://www.mdpi.com/2073-4441/11/11/2362 (see also the references).

Be critical with your data and methodology, to use the stronger information you get.

Finally, don let this information down. Fish and fishways need it.

Good luck!